biochemistry/biotechnology

genetically modified soya bean, real-time PCR, traceability, soya bean protein concentrate, soya bean protein isolate

**Author for correspondence:**
Fusheng Chen
e-mail: fushengc@haut.edu.cn

This article has been edited by the Royal Society of Chemistry, including the commissioning, peer review process and editorial aspects up to the point of acceptance.

# Monitoring and traceability of genetically modified soya bean event GTS 40-3-2 during soya bean protein concentrate and isolate preparation

## Yan Du, Fusheng Chen, Chen Chen and Kunlun Liu

College of Food Science and Technology, Henan University of Technology, 100, Lianhua Street, High-tech, Zhengzhou 450001, Henan, People's Republic of China

YD, 0000-0003-4123-8720; FC, 0000-0002-8201-1234; CC, 0000-0001-5914-228X

To evaluate DNA fragmentation and GMO quantification during soya bean protein concentrate and isolate preparation, genetically modified soya bean event GTS 40-3-2 (Roundup Ready™ soya bean, RRS) was blended with conventional soya beans at mass percentages of 0.9%, 2%, 3%, 5% and 10%. Qualitative PCR and real-time PCR were used to monitor the taxon-specific *lectin* and exogenous *cp4 epsps* target levels in all of the main products and by-products, which has practical significance for RRS labelling threshold and traceability. Along the preparation chain, the majority of DNA was distributed in main products, and the DNA degradation was noticed. From a holistic perspective, the *lectin* target degraded more than *cp4 epsps* target during both of the two soya bean proteins preparations. Therefore, the transgenic contents in the final protein products were higher than the actual mass percentages of RRS in raw materials. Our results are beneficial to the improvement of GMO labelling legislation and the protection of consumer rights.

## 1. Introduction

Since the genetically modified organism (GMO) was commercially released, the cultivation of GMOs and their application in food and feed products has become increasingly widespread [1]. According to the International Service for the Acquisition of Agri-biotech Applications (ISAAA), 191.7 million hectares (a 113-fold

increase since 1996) of GMOs were cultivated globally in 2018, considerably improving food availability by increasing yields and reducing losses [2].

With the consumption of GMO products, transgenic DNA and protein originating from GMO are likely to enter the human food chain [3], increasing the consumer concern about the potential hazards of GMO products [4]. To regulate the circulation of GMO products in the international market and help consumers make informed choices about foods that may contain GMO, many countries have established rules regarding the labelling of GMO products [5,6]. The rules in Japan, Thailand, Taiwan, South Africa and Indonesia state that any food product should be clearly labelled if it contains 5% or more GMO components [7]. The corresponding threshold values are 3% in South Korea [8] and 0.9% in the European Union and Russia [9].

However, the accurate labelling of GMO products is difficult, mainly because of the degradation of food components, like DNA, during food processing [3]. Furthermore, the different extents of degradation of taxon-specific and exogenous targets inevitably distort the determination of GMO contents in processed foods [10]. Hence, the traceability, i.e. the ability to track GMO products at all stages of their entry into the market, is vital for accurate GMO labelling and GMO product transparency [11].

Previous studies on GMO monitoring and traceability have chiefly concentrated on tofu [10], bread [12], soya bean milk [13], rice noodles [3] and other food processing chains [14–16]. However, the DNA degradation and GMO quantification during the preparation of soya bean protein were less discussed, though soya bean is the most cultivated GM crop around the world [17], and the soya bean protein is a superior plant protein that features commonly in our daily diets [18].

Therefore, in this study, raw soya beans with GM soya bean event GTS 40-3-2 (Roundup Ready™ soya bean, RRS) proportions of 0.9%, 2%, 3%, 5% and 10% (w/w) were used to prepare two major forms of soya bean protein, i.e. soya bean protein concentrate (SPC) and soya bean protein isolate (SPI). The variations in DNA fragment distribution and RRS proportion were assessed systematically. We hope that our results will help to inform consumers, maintain their rights to choose and promote the sustainable development of GMO-derived products.

# 2. Materials and methods

## 2.1. Materials

To construct standard curves, soya bean flour certified reference materials (CRMs) containing 0%, 0.1%, 1%, 10% and 100% (w/w) RRS were developed by the Institute for Research Materials and Measurements (Geel, Belgium) and purchased from Shanghai ZZBio Co., Ltd (Shanghai, China). Equal amount of CRM containing 0% and 100% (w/w) RRS was thoroughly mixed to obtain a further CRM containing 50% (w/w) RRS [3].

To prepare SPC and SPI, conventional soya bean and RRS were supported by the Institute of Crop Sciences, Chinese Academy of Agricultural Sciences (Beijing, China). Then, five groups of raw soya beans with RRS proportions of 0.9%, 2%, 3%, 5% and 10% (w/w) were obtained by mixing appropriate masses of RRS and conventional soya bean thoroughly. For instance, raw soya bean with 10% (w/w) RRS was prepared by mixing 20.00 g of RRS with 180.00 g of conventional soya bean. Five replicates of each group were prepared, one of which was used for RRS quantification, two for SPC preparation and two for SPI preparation.

## 2.2. Soya bean protein concentrate preparation

Figure 1 illustrates the SPC preparation procedures, which were accomplished in our laboratory through ethanol extraction method. First, 200 g of raw soya bean was subjected to dehulling with a JGMJ8098 mini huller (Jiading Cereals and Oils Instrument Co. Ltd, Shanghai, China). Then, by using the FW-100 high-speed smashing machine (Ever Bright Medical Treatment Instrument Co. Ltd, Beijing, China), all of the soya bean kernels were smashed into flour to pass through the 60-mesh sieve. Defatting was done by three repeated n-hexane extractions at a solvent-to-flour ratio of 5 : 1 (v/w), a stirring temperature of 35°C, and a stirring time of 2 h. After each extraction, the slurry was centrifuged at 3000×g for 20 min. Then, the supernatant was carefully transferred to a round-bottom flask and the organic solvent was evaporated at 45°C by using a RE100-S rotary evaporator (DLAB Scientific Co. Ltd, Beijing, China). The precipitate was dried overnight in a fume hood to obtain the defatted soya bean kernel.

Thereafter, 65%, 85%, 70% and 70% ethanol were applied for the first, second, third and fourth ethanol extraction, respectively, at the same solid–liquid ratio of 1 : 10 (w/v). During the first ethanol

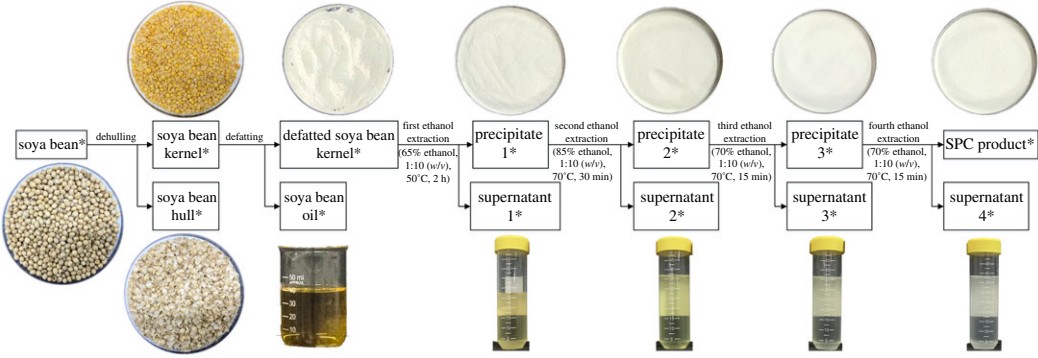

**Figure 1.** Flow chart of SPC preparation.

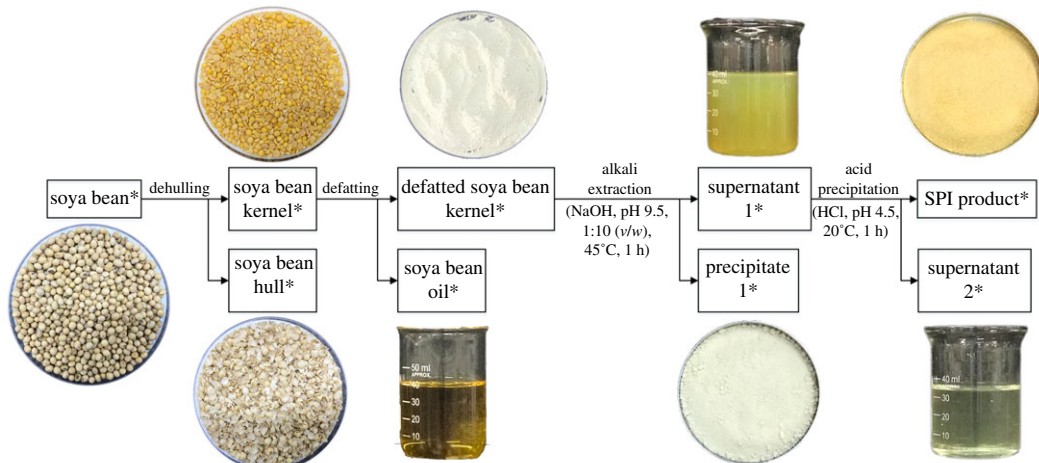

**Figure 2.** Flow chart of SPI preparation.

extraction, the mixture was stirred at 50°C for 2 h. The stirring temperature for the second, third and fourth ethanol extractions was 70°C, and the stirring times were 30, 15 and 15 min, respectively. After each ethanol extraction, the slurry was centrifuged at 3000×$g$ for 20 min to produce the corresponding SPC precipitates and SPC supernatants, which were then lyophilized before DNA extraction. Along the preparation chain, all of the main products and by-products were collected as indicated with asterisks in figure 1 and stored at 4°C for further study.

## 2.3. Soya bean protein isolate preparation

As shown in figure 2, the alkali extraction–acid precipitation method was applied to prepare SPI, starting with 200g of raw soya bean material and following the same procedures of dehulling and defatting as those used for SPC preparation. Then, deionized water was added to the defatted soya bean kernel at a ratio of 10 : 1 (v/w), and 2 M NaOH was used to adjust the pH of the mixture to 9.5. Then, the mixture was stirred at 45°C for 1 h and centrifuged at 3000×$g$ for 20 min to obtain SPI supernatant 1 and SPI precipitate 1. Next, 2 M HCl was used to adjust the pH of the SPI supernatant 1 to 4.5. After 1 h storage at 20°C, the mixture was centrifuged at 3000×$g$ for 20 min to obtain SPI supernatant 2 and SPI product. Finally, the SPI supernatant 1, SPI supernatant 2, SPI precipitate 1 and SPI product were lyophilized before DNA extraction. Along the preparation chain, all of the main products and by-products were collected as indicated with asterisks in figure 2 and stored at 4°C for further study.

## 2.4. DNA extraction

DNAs from solid samples were extracted by using a DNeasy Plant Mini Kit (Qiagen, Hilden, Germany) on the basis of the manufacturer's handbook with slight modifications. The solid samples were smashed and homogenized into fine powders to pass through a 60-mesh sieve. Then, 800 µl (instead of 400 µl)

buffer AP1 was added to 100 mg of powdered sample to continue the subsequent extraction steps. An Oil DNA Extraction Kit (Dingguo Changsheng Biotechnology Co. Ltd, Beijing, China) was applied for extracting DNA from soya bean oil, and the extractions were performed according to the manufacturer's procedures without modification. Each extraction was performed at least three times.

A Nanodrop 2000 (Thermo) was used to assess the mass concentration and purity of DNA according to the UV absorption at 260, 280 and 230 nm. Afterwards, the DNA extracts were diluted with DNase/RNase-Free Water (R1600, Solarbio Life Sciences, Beijing, China), sub-packed in multiple tubes, and kept at −20°C until further analysis.

## 2.5. Qualitative PCR amplification and gel analysis

Five and seven pairs of primers, synthesized by Sangon Biotechnology (Shanghai, China), were used to amplify the fragments within the soya bean taxon-specific *lectin* (GenBank accession number: K00821) and exogenous *cp4 epsps* (GenBank accession number: AB209952) gene, respectively. Detailed information on the primers is presented in electronic supplementary material, table S1.

Qualitative PCR (25 µl) contained 12.5 µl 2×Premix Taq (TaKaRa Bio Inc., Beijing, China), 0.4 µM each primer and 50 ng template DNA. The amplifications were performed using a 96-well T100 Thermal Cycler (Bio-Rad Laboratories Inc., Hercules, USA) under the conditions at 95°C for 5 min; 35 cycles of 95°C for 30 s, 52–63.1°C (electronic supplementary material, table S1) for 30 s, and 72°C for 9–90 s (electronic supplementary material, table S1); 72°C for 10 min. DNAs from all samples collected during the preparations were amplified. Blank control with no template DNA, as well as negative control and positive control were involved in each run.

The PCR products were monitored through electrophoresis at 120 V with 2% agarose gel containing 4S Green Plus Nucleic Acid Stain (0.1 µl ml$^{-1}$, Sangon Biotechnology). Marker I and D2000 Marker (MD101 and MD114, Tiangen Biotech Co. Ltd, Beijing, China) were used as size references. Visualization of the gels was performed with a Tanon 2500 Gel Imaging System (Tanon Science & Technology Co. Ltd, Shanghai, China).

## 2.6. Real-time quantitative PCR (qPCR)

qPCR was carried out on a 96-well QuantStudio$^{TM}$ 3 Real-time PCR System (Thermo). Primers for amplifying the 81 bp of *lectin* target (Lectin-F/R) and 83 bp of *cp4 epsps* target (RRS-F/R) were used as previously described [19] and synthesized by Sangon Biotechnology (Shanghai, China).

Each reaction (20 µl) consisted of 10 µl 2× PowerUp$^{TM}$ SYBR$^{TM}$ Green Master Mix (Thermo), 0.3 µM each primer, and 50 ng template DNA. In each run, the taxon-specific and exogenous target reactions with DNA from 0.1%, 1%, 10%, 50% and 100% (w/w) RRS CRM were done in separate tubes for standard curves construction. Meanwhile, non-template DNA controls were set up with DNase/RNase-Free Water (Solarbio) to ensure the absence of contamination.

The qPCR protocol contained an Uracil DNA glycosylase activation at 50°C for 2 min, Dual-Lock$^{TM}$ Taq DNA polymerase activation at 95°C for 2 min, and 40 cycles of denaturation at 95°C for 15 s and annealing at 60°C for 1 min. The fluorescence signals were recorded once each cycle after the annealing step. Four replicates were carried out for each sample.

## 2.7. Determination of RRS concentrations

After completing the run, the quantification cycle (Cq) values [20] of each target were obtained and data were analysed by the QuantStudio$^{TM}$ Design & Analysis Software (Thermo). The ΔCq value, equivalent to the difference between the Cq value of the *cp4 epsps* target and that of the *lectin* target, was calculated. Because of the linear relationship between log(RRS%) and ΔCq value of each sample, the RRS quantification in trial samples was completed by interpolation on a standard regression curve of ΔCq values generated from DNA extracts of known RRS concentration (%) [13].

## 2.8. Statistical analysis

The experimental results were compared by one-way analysis of variance. All data were expressed as means ± s.d. (standard deviation).

**Table 1.** Average mass concentration and purity of DNA per product collected during SPC and SPI preparation.

| sample name | mass concentration of DNA (ng μl$^{-1}$) | DNA purity ($A_{260}/A_{280}$) | DNA purity ($A_{260}/A_{230}$) |
|---|---|---|---|
| Soya bean | 117.53 ± 10.54 | 1.88 ± 0.04 | 2.58 ± 0.29 |
| Soya bean hull | 78.73 ± 13.37 | 1.88 ± 0.05 | 2.68 ± 0.21 |
| Soya bean kernel | 118.93 ± 24.35 | 1.87 ± 0.05 | 2.70 ± 0.34 |
| Defatted soya bean kernel | 147.53 ± 23.28 | 1.87 ± 0.05 | 2.56 ± 0.48 |
| SPC precipitate 1 | 71.93 ± 13.14 | 1.88 ± 0.04 | 2.60 ± 0.23 |
| SPC precipitate 2 | 69.80 ± 17.69 | 1.87 ± 0.05 | 2.83 ± 1.01 |
| SPC precipitate 3 | 65.07 ± 21.11 | 1.85 ± 0.06 | 2.74 ± 0.66 |
| SPC product | 57.20 ± 16.34 | 1.88 ± 0.05 | 2.46 ± 0.42 |
| SPC supernatant 1 | 12.47 ± 4.16 | 2.48 ± 0.23 | 0.55 ± 0.27 |
| SPC supernatant 2 | 12.60 ± 3.38 | 2.23 ± 0.45 | 0.91 ± 0.42 |
| SPC supernatant 3 | 11.93 ± 5.40 | 2.16 ± 0.40 | 0.61 ± 0.35 |
| SPC supernatant 4 | 10.07 ± 2.76 | 1.44 ± 0.16 | 0.74 ± 0.39 |
| SPI supernatant 1 | 128.33 ± 22.04 | 1.89 ± 0.06 | 2.67 ± 0.31 |
| SPI product | 71.87 ± 23.35 | 1.90 ± 0.04 | 2.53 ± 0.25 |
| SPI precipitate 1 | 68.47 ± 14.48 | 1.89 ± 0.05 | 2.34 ± 0.36 |
| SPI supernatant 2 | 9.67 ± 1.68 | 1.66 ± 0.06 | 1.00 ± 0.32 |
| Soya bean oil | 20.80 ± 4.87 | 1.88 ± 0.05 | 1.23 ± 0.21 |

# 3. Results and discussion

## 3.1. General spectrophotometric analysis of DNA extracts

Two commercial DNA extraction kits were applied to obtain DNA from the main products and by-products after each step of SPC and SPI preparation. The mass concentrations of all DNA extracts were measured through spectrophotometry, and the average values of all the RRS levels per product were calculated accordingly. As shown in table 1, the average mass concentration ranged from 9.67 ± 1.68 ng μl$^{-1}$ in SPI supernatant 2 to 147.53 ± 23.28 ng μl$^{-1}$ in defatted soya bean kernel.

The purities of DNA extracts are generally reflected by their $A_{260}/A_{280}$ and $A_{260}/A_{230}$ ratios [21]. When the $A_{260}/A_{280}$ ratio is around 1.5–2.0 and the $A_{260}/A_{230}$ ratio exceeds 1.7, the DNA is deemed pure without any contamination by compounds like proteins, carbohydrates and phenols [22]. Therefore, most of the samples in this research exhibited adequate purity, while the $A_{260}/A_{280}$ and $A_{260}/A_{230}$ ratios were out of optimal range for some samples, such as SPC supernatants 1–4.

## 3.2. DNA fragmentation along soya bean protein concentrate preparation process

The taxon-specific *lectin* targets with lengths of 60, 201, 414, 836 and 1487 bp were amplified in each product collected along SPC preparation. As illustrated in figure 3, fragments of 60–1487 bp were observed in all of the main products and one of the by-products (soya bean hull). However, the brightnesses of 1487 bp fragments for SPC precipitates 1–3, SPC product, and soya bean hull were not only poorer than those of 836 bp (and below) in the corresponding products, but also poorer than the brightness of 1487 bp in soya bean, dehulled soya bean and defatted soya bean kernel. In addition, for SPC supernatants and soya bean oil, only weaker bands of 60–201 bp and 60–414 bp were visible, respectively, compared with those in other samples.

These phenomena manifested that the *lectin* target of 1487 bp was distributed more in soya bean kernel than in soya bean hull after dehulling. Upon defatting, the majority of the *lectin* target was transferred to defatted soya bean kernel, while only a few fragments of 60–414 bp were present in soya bean oil. This

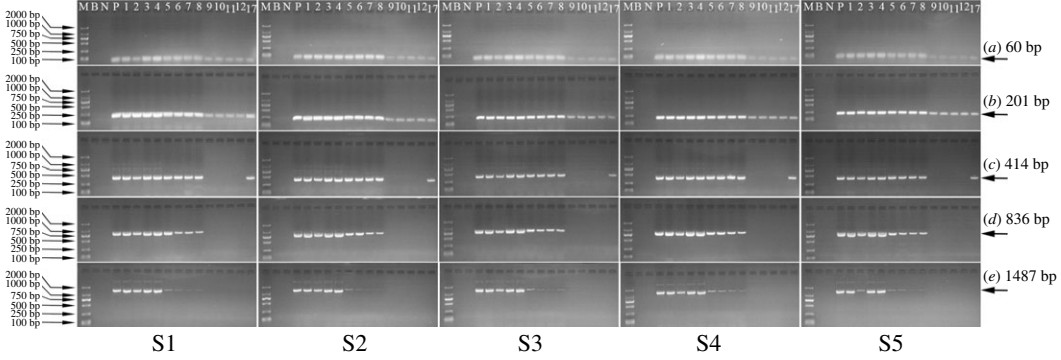

**Figure 3.** Qualitative PCR amplification of the *lectin* target during SPC preparation starting from soya bean with (S1) 0.9%, (S2) 2%, (S3) 3%, (S4) 5% and (S5) 10% RRS. Lane (M) D2000 Marker, (B) blank, (N) maize (negative control), (P) 100% RRS CRM (positive control), (1) raw soya bean material, (2) soya bean hull, (3) soya bean kernel, (4) defatted soya bean kernel, (5) SPC precipitate 1, (6) SPC precipitate 2, (7) SPC precipitate 3, (8) SPC product, (9) SPC supernatant 1, (10) SPC supernatant 2, (11) SPC supernatant 3, (12) SPC supernatant 4, (17) soya bean oil.

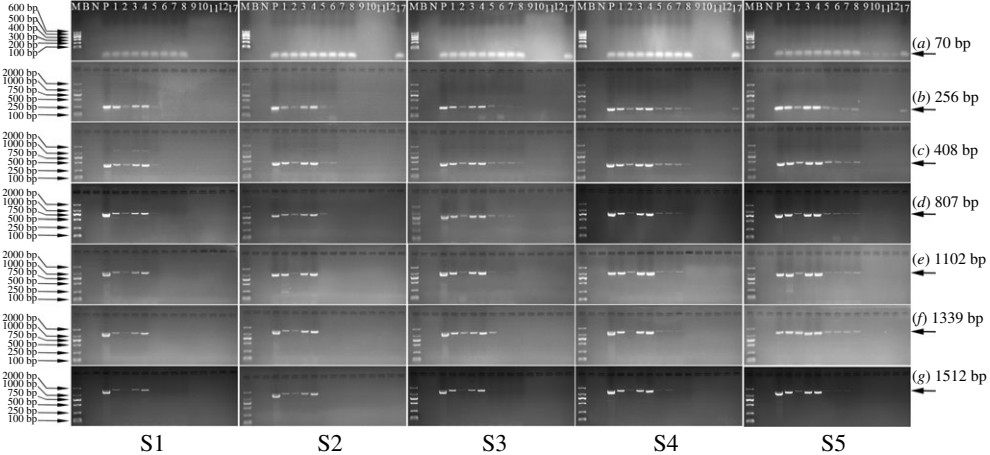

**Figure 4.** Qualitative PCR amplification of the *cp4 epsps* target during SPC preparation starting from soya bean with (S1) 0.9%, (S2) 2%, (S3) 3%, (S4) 5% and (S5) 10% RRS. Lane (M) D2000 Marker, (B) blank, (N) 0% RRS CRM (negative control), (P) 100% RRS CRM (positive control), (1) raw soya bean material, (2) soya bean hull, (3) soya bean kernel, (4) defatted soya bean kernel, (5) SPC precipitate 1, (6) SPC precipitate 2, (7) SPC precipitate 3, (8) SPC product, (9) SPC supernatant 1, (10) SPC supernatant 2, (11) SPC supernatant 3, (12) SPC supernatant 4, (17) soya bean oil.

discovery was in agreement with that of Costa *et al.* [23], who reported the difficulty in amplifying DNA distributed in soya bean oil on account of low DNA integrity and PCR inhibitors' existence.

Ethanol extraction was the most critical step for removing the *lectin* target during SPC preparation, because there were weak bands of 60–201 bp and no bands of 414–1487 bp in SPC supernatants. Moreover, the fragments of 836 and 1487 bp were found to be largely degraded after the first ethanol extraction, and almost no fragment of 1487 bp could be observed after the fourth ethanol extraction. Thus, our results confirmed previous findings that DNA in botanicals could be either heavily degraded or completely eliminated upon solvent extraction [24].

Sizes of the exogenous DNA segments were also qualitatively analysed after the preparation of SPC from raw soya bean with 0.9%, 2%, 3%, 5% and 10% (w/w) RRS, respectively (figure 4). Again, the majority of the *cp4 epsps* target was distributed in the main products, while only a few or even no fragments were distributed in the by-products. Maybe due to the lower relative content of the *cp4 epsps* target compared with the *lectin* target, the degradation of the *cp4 epsps* target was more apparent as affected by ethanol extraction. With increasing ethanol extraction times, the amplifiable *cp4 epsps* target fragments disappeared gradually from long to short.

Besides, as verified in electronic supplementary material, figure S1, the bands for the *cp4 epsps* target with a specific size (70, 256, 408, 807, 1102, 1339 or 1512 bp) became weaker as the RRS content decreased under the same PCR conditions with same amount of template DNA. Hence, it was seen that the size

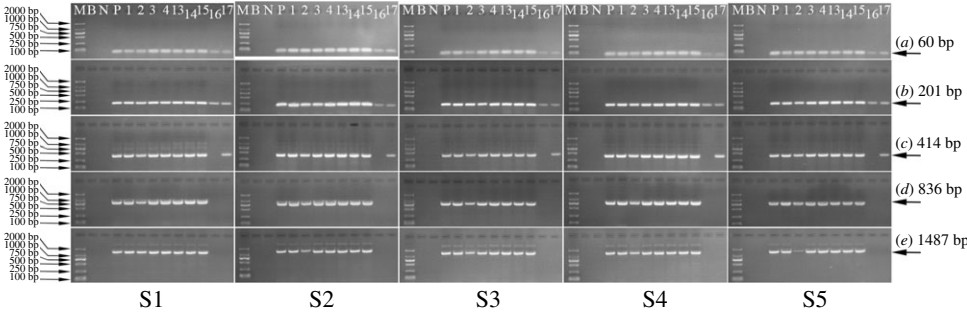

**Figure 5.** Qualitative PCR amplification of the *lectin* target during SPI preparation starting from soya bean with (S1) 0.9%, (S2) 2%, (S3) 3%, (S4) 5% and (S5) 10% RRS. Lane (M) D2000 Marker, (B) blank, (N) maize (negative control), (P) 100% RRS CRM (positive control), (1) raw soya bean material, (2) soya bean hull, (3) soya bean kernel, (4) defatted soya bean kernel, (13) SPI supernatant 1, (14) SPI product, (15) SPI precipitate 1, (16) SPI supernatant 2, (17) soya bean oil.

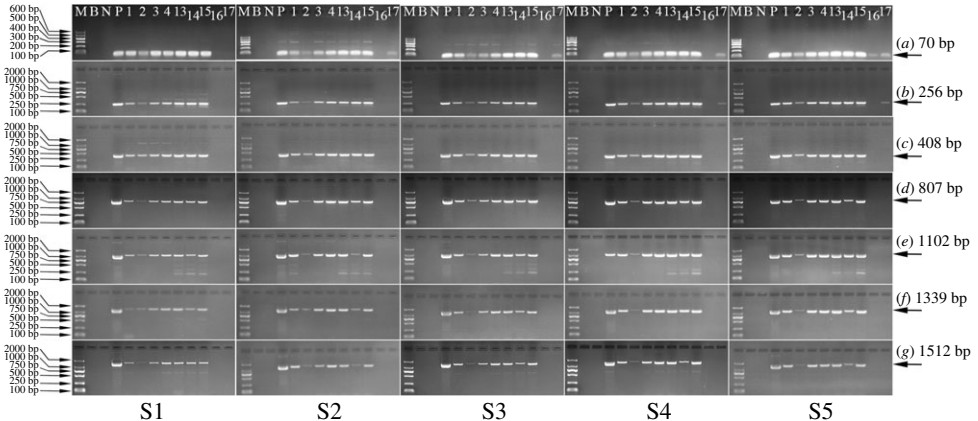

**Figure 6.** Qualitative PCR amplification of the *cp4 epsps* target during SPI preparation starting from soya bean with (S1) 0.9%, (S2) 2%, (S3) 3%, (S4) 5% and (S5) 10% RRS. Lane (M) D2000 Marker, (B) blank, (N) 0% RRS CRM (negative control), (P) 100% RRS CRM (positive control), (1) raw soya bean material, (2) soya bean hull, (3) soya bean kernel, (4) defatted soya bean kernel, (13) SPI supernatant 1, (14) SPI product, (15) SPI precipitate 1, (16) SPI supernatant 2, (17) soya bean oil.

ranges of the amplifiable *cp4 epsps* target in the main products obtained after the first to fourth ethanol extractions differed when raw soya beans with different RRS percentages were used to prepare SPC. With the decrease of RRS percentage in raw soya beans, the *cp4 epsps* target fragments in SPC precipitate 1, SPC precipitate 2, SPC precipitate 3 and SPC product disappeared gradually from long to short, respectively. Typically, no fragments of 1102–1512 bp were observed in SPC precipitate 1 started with 0.9% and 2% (w/w) RRS. In SPC precipitate 2, there were no fragments of 256–1512 bp for 0.9% (w/w) RRS, 807–1512 bp for 2% (w/w) RRS and 1339–1512 bp for 3% (w/w) RRS, respectively. No further degradation was monitored in SPC precipitate 3 started with 0.9% and 3% (w/w) RRS, while 256–408 bp and 1512 bp of the *cp4 epsps* target were degraded to be undetectable in that started with 2% and 5% (w/w) RRS, respectively. The fourth ethanol extraction caused no additional degradation of the *cp4 epsps* target in SPC products started with 0.9% and 2% (w/w) RRS. Yet, the fragments of 807–1512 bp were absent in SPC products started with 3% and 5% (w/w) RRS.

## 3.3. DNA fragmentation along soya bean protein isolate preparation process

Different fragments of the taxon-specific *lectin* (figure 5) and exogenous *cp4 epsps* target (figure 6) were amplified in samples taken during SPI preparation using 0.9%–10% (w/w) RRS. Since the dehulling and defatting operations were similar to those used in SPC preparation, the distribution and degradation patterns of both targets were similar in these two steps.

Alkali extraction and acid precipitation were the other two primary procedures for preparing SPI. For the reason that numerous PCR products of 60–1487 bp still existed in SPI supernatant 1, SPI precipitate 1 and SPI product, neither the alkali nor the acid step was perceived to have much of an effect on the

degradation of the *lectin* target, agreeing with Bauer *et al.* [25]. Alkali extraction at pH 9.5 caused no obvious degradation of the *cp4 epsps* target either. However, owing to the low content of RRS (0.9%–10%, w/w), the degradation of the *cp4 epsps* target during acid precipitation was visualized. As indicated in figure 6, the fragments between 807 bp and 1512 bp in SPI products were noticeably less abundant than those in SPI supernatant 1. The reason would be attributed to the depurination, hydrolysis and/or enzymatic degradation of DNA in acid environments [26].

What is more, the distributions of both targets differed in the main product and by-product after incubating under acidic conditions. Limited amounts of both targets were distributed in the by-product. For instance, only a few short fragments, like 60 and 201 bp, for the *lectin* target were detectable in SPI supernatant 2. Additionally, only the *cp4 epsps* target with size of 70 bp was observed in SPI supernatant 2 started with 10% (w/w) RRS. Besides, there was no fragment (≥70 bp) appearing in SPI supernatant 2 started with 0.9%, 2%, 3% or 5% (w/w) RRS.

## 3.4. Construction of standard curves

The PCR target sizes are vital for GMO quantification in food products [27]. Aiming at acquiring analogous PCR efficiencies [28], qPCR assays, that amplified small and comparable fragments (around 80 bp) of the taxon-specific and exogenous targets, were employed to assess the relative content variations of RRS during soya bean protein preparation. A series of CRMs containing 0.1%–100% (w/w) RRS were used for standard curves calibration. As shown in electronic supplementary material, figure S2, all of the standard curves applied in this research revealed high linearity between the log(RRS%) and ∆Cq (Cq *cp4 epsps*-Cq *lectin*) with correlation coefficients ($R^2$) higher than 0.99 [29].

## 3.5. Monitoring of RRS proportions along SPC preparation process

Five groups of soya beans with 0.9%, 2%, 3%, 5% and 10% (w/w) RRS were used as raw materials to produce SPC. The RRS proportions in these soya beans as measured by qPCR were 0.91%, 2.06%, 3.04%, 5.00% and 10.06%, respectively (table 2), which accords with their original RRS mass percentages. This confirmed that there were little differences existing in the genome/weight ratios of the five levels of raw soya beans without any processes [13].

However, dehulling altered the RRS proportions in soya bean hull and soya bean kernel. Compared with the RRS proportion in raw soya bean, that in soya bean hull increased by 76.84%–104.00%, while that in soya bean kernel decreased by 2.20%–17.79%. Among the five groups of samples, the RRS proportion in soya bean kernel prepared with 5% and 10% (w/w) RRS declined significantly ($p < 0.05$) to 4.28% and 8.27%, respectively. This reduction of RRS content in the main product after mechanical manipulation was similar to those observed in previous studies, indicating that more damage was suffered by the exogenous target than by the *lectin* target [7]. Then, after defatting, the RRS proportions in soya bean oil and defatted soya bean kernel were higher than those in raw soya bean and soya bean kernel. This finding was consistent with that reported previously [23] and may be owing to the higher stability of the *cp4 epsps* target under n-hexane treatment.

Ethanol extraction had evident impacts on the degradation of the taxon-specific *lectin* and exogenous *cp4 epsps* target, and the majority of both targets was distributed in the main products after each procedure (figures 3 and 4). When the defatted soya bean kernel was stirred in 65% (1:10, w/v) ethanol at 50°C for 2 h, the RRS proportion in SPC precipitate 1 derived from each group of raw soya beans continued to increase, and the increments in groups prepared with 3% and 5% (w/w) RRS were significant ($p < 0.05$) (table 2). Therefore, the *lectin* target was considered to be more damaged than the *cp4 epsps* target during the first ethanol extraction. Thereafter, probably due to the limited and varying degrees of degradation suffered by the taxon-specific and exogenous targets, the RRS percentages fluctuated with no significant difference in the main products after the second to fourth ethanol extractions.

When it came to the by-products gained after four times of ethanol extractions, no amplification of the *cp4 epsps* target was observed in SPC supernatant 1–4 derived from 0.9% and 2% (w/w) RRS, or SPC supernatant 3–4 derived from 3% and 5% (w/w) RRS. This phenomenon confirmed the qualitative PCR results in figure 4 and was probably caused by the low concentration of the *cp4 epsps* target and inhibitors present in DNA extracts [23]. Beyond that, positive signals were amplified in SPC supernatants 1–2 derived from 3% and 5% (w/w) RRS, and SPC supernatants 1–4 derived from 10% (w/w) RRS.

**Table 2.** Quantitative results of RRS proportions (%) in samples during SPC preparation.

| sample name | content of *cp4 epsps* target, RRS (%) | | | | |
| --- | --- | --- | --- | --- | --- |
| | S1 | S2 | S3 | S4 | S5 |
| Soya bean | 0.91 ± 0.03[a] (4/4)[#] | 2.06 ± 0.14[ab] (4/4) | 3.04 ± 0.37[ab] (4/4) | 5.00 ± 0.16[b] (4/4) | 10.06 ± 0.38[b] (4/4) |
| Soya bean hull | 1.74 ± 0.48* (4/4) | 3.80 ± 0.23* (4/4) | 6.19 ± 0.86* (4/4) | 9.55 ± 1.02* (4/4) | 17.79 ± 0.41* (4/4) |
| Soya bean kernel | 0.89 ± 0.09[a] (4/4) | 1.88 ± 0.07[a] (4/4) | 2.82 ± 0.18[a] (4/4) | 4.28 ± 0.20[a] (4/4) | 8.27 ± 0.14[a] (4/4) |
| Defatted soya bean kernel | 1.15 ± 0.05[ab] (4/4) | 2.52 ± 0.09 [abc] (4/4) | 3.45 ± 0.17 [bc] (4/4) | 5.21 ± 0.24 [b] (4/4) | 10.60 ± 1.40[bc] (4/4) |
| SPC precipitate 1 | 1.52 ± 0.47[b] (4/4) | 2.87 ± 0.74[c] (4/4) | 4.45 ± 0.68[d] (4/4) | 7.08 ± 0.40[c] (4/4) | 11.93 ± 0.46[c] (4/4) |
| SPC precipitate 2 | 1.16 ± 0.43[ab] (4/4) | 2.65 ± 0.31[bc] (4/4) | 4.11 ± 0.42[d] (4/4) | 6.66 ± 0.72[c] (4/4) | 11.53 ± 0.94[bc] (4/4) |
| SPC precipitate 3 | 1.25 ± 0.27[ab] (4/4) | 2.61 ± 0.75[bc] (4/4) | 4.06 ± 0.41[d] (4/4) | 6.49 ± 0.40[c] (4/4) | 11.18 ± 1.45[bc] (4/4) |
| SPC product | 1.19 ± 0.28[ab] (4/4) | 2.68 ± 0.28[bc] (4/4) | 3.96 ± 0.24[cd] (4/4) | 6.53 ± 0.46[c] (4/4) | 11.41 ± 0.92[bc] (4/4) |
| SPC supernatant 1 | ND* | ND* | 2.38 ± 0.89* (2/4) | 5.43 ± 0.72* (3/4) | 9.03 ± 3.51* (4/4) |
| SPC supernatant 2 | ND* | ND* | 2.77 ± 0.90* (2/4) | 4.57 ± 1.48* (2/4) | 7.41 ± 0.95* (2/4) |
| SPC supernatant 3 | ND* | ND* | ND* | ND* | 15.09 ± 8.63* (2/4) |
| SPC supernatant 4 | ND* | ND* | ND* | ND* | 9.27 ± 0.93* (2/4) |
| Soya bean oil | 2.81 ± 1.73* (4/4) | 9.45 ± 0.89* (4/4) | 6.16 ± 1.34* (4/4) | 7.79 ± 1.46* (4/4) | 14.78 ± 2.62* (4/4) |

S1, S2, S3, S4, S5—Raw soya bean materials with 0.9%, 2%, 3%, 5%, 10% (w/w) RRS%, respectively.

The different superscript lowercase letters in each column indicate significant difference ($p < 0.05$).

ND—None detected under the conditions used.

*Quantitative results of by-products that do not participate in the significance analysis.

#Positive replicates/total of replicates.

Overall, the *lectin* target was found to suffer more damage than the *cp4 epsps* target during SPC preparation. Consequently, the RRS proportions in the final SPC products were higher than those in raw soya beans for the five experimental groups. In the European Union, GM products must be labelled if they contain more than 0.9% exogenous components [9]. Therefore, although the RRS percentage in raw material is below this threshold, labelling may still be required for SPC.

## 3.6. Monitoring of roundup ready™ soya bean proportions along soya bean protein isolate preparation process

The RRS proportions in all samples throughout SPI preparation are displayed in table 3. On account of the same dehulling and defatting procedures being used for both SPI and SPC preparation, the variations of RRS proportions in these procedures were the same as those discussed in §3.5. Thereafter, the defatted

**Table 3.** Quantitative results of RRS proportions (%) in samples during SPI preparation.

| samples | content of *cp4 epsps* target, RRS (%) | | | | |
| --- | --- | --- | --- | --- | --- |
| | S1 | S2 | S3 | S4 | S5 |
| Soya bean | 0.91 ± 0.03[a] (4/4)[#] | 2.06 ± 0.14[ab] (4/4) | 3.04 ± 0.37[ab] (4/4) | 5.00 ± 0.16[b] (4/4) | 10.06 ± 0.38[b] (4/4) |
| Soya bean hull | 1.74 ± 0.48* (4/4) | 3.81 ± 0.23* (4/4) | 6.19 ± 0.86* (4/4) | 9.55 ± 1.02* (4/4) | 17.79 ± 0.41* (4/4) |
| Soya bean kernel | 0.89 ± 0.09[a] (4/4) | 1.88 ± 0.07[a] (4/4) | 2.82 ± 0.18[a] (4/4) | 4.28 ± 0.20[a] (4/4) | 8.27 ± 0.14[a] (4/4) |
| Defatted soya bean kernel | 1.15 ± 0.05[bc] (4/4) | 2.52 ± 0.09[cd] (4/4) | 3.45 ± 0.17[cd] (4/4) | 5.21 ± 0.24[bc] (4/4) | 10.60 ± 1.40[b] (4/4) |
| SPI supernatant 1 | 1.23 ± 0.10[c] (4/4) | 2.59 ± 0.40[d] (4/4) | 3.59 ± 0.02[d] (4/4) | 5.73 ± 0.19[c] (4/4) | 10.68 ± 0.81[b] (4/4) |
| SPI product | 1.12 ± 0.03[b] (4/4) | 2.25 ± 0.17[bc] (4/4) | 3.22 ± 0.07[bc] (4/4) | 5.71 ± 0.79[c] (4/4) | 10.10 ± 0.07[b] (4/4) |
| SPI precipitate 1 | 1.09 ± 0.10* (4/4) | 2.14 ± 0.19* (4/4) | 3.36 ± 0.16* (4/4) | 5.18 ± 0.06* (4/4) | 10.28 ± 0.66* (4/4) |
| SPI supernatant 2 | ND* | ND* | ND* | ND* | 30.91 ± 11.96* (2/4) |
| Soya bean oil | 2.81 ± 1.73* (4/4) | 9.45 ± 0.89* (4/4) | 6.16 ± 1.34* (4/4) | 7.79 ± 1.46* (4/4) | 14.78 ± 2.62* (4/4) |

S1, S2, S3, S4, S5—Raw soya bean materials with 0.9%, 2%, 3%, 5%, 10% (w/w) RRS%, respectively.
The different superscript lowercase letters in each column indicate significant difference ($p < 0.05$).
ND—None detected under the conditions used.
*Quantitative results of by-products that do not participate in the significance analysis.
#Positive replicates/total of replicates.

soya bean kernel was stirred at 45°C for 1 h in an alkaline environment. Since alkali extraction had no remarkable impact on degradation of the *lectin* and *cp4 epsps* targets (figures 5 and 6), the RRS percentage of each group varied to some extent with no significant difference. However, in the acid precipitation process, the RRS proportions in SPI products (1.12%–3.22%) derived from 0.9% to 3% (*w/w*) RRS were significantly lower ($p < 0.05$) in SPI supernatant 1 (1.23%–3.59%). Hence, low pH conditions induced more distinct degradation in the *cp4 epsps* target than in the *lectin* target, as indicated in §3.3. Nonetheless, from an overall point of view, the RRS proportions in SPI products were elevated compared with those in raw soya beans. Therefore, the *cp4 epsps* target was more stable than the *lectin* target throughout the SPI preparation.

DNA fragmentation and exogenous target level variation along food processing procedures have been of interests for many years. Because shorter DNA segments tend to be more resistant to degradation than longer ones, small taxon-specific and exogenous targets with approximate size are commonly used for GMO quantification [19], as in this research. On this premise, we discovered the distortions of RRS proportions in products obtained during soya bean protein preparation, which has barely been focused on before.

At present, there are several reports concerning other food preparation chains. For instance, Guan *et al*. [27] stated that the GMO content in Bt cottonseed meal decreased with increasing treatment temperature. Besides, after oven baking, the concentrations of MON810 maize in maize breads (44%–67%) were found to be lower than that in raw material (103%) [12]. Furthermore, an increment in RRS concentration from 0.44% to 1.15% was observed after CaSO$_4$ addition during soya bean curd processing [10].

The physical and chemical treatments involved in food processing are generally considered to cause different degrees of degradation between taxon-specific and exogenous DNA, affecting GMO quantification as a consequence [30]. However, the exact mechanisms leading to the stability discrepancy

between different DNA targets under specific conditions has not been fully established and needs further investigation. Possible affecting factors are differences in the guanine-cytosine content between each target and different characteristics of the ingredients in food matrix [3,19].

# 4. Conclusion

In this research, a scale of raw soya beans containing 0.9%–10% (w/w) RRS were used for preparing SPC and SPI. The state of the taxon-specific and exogenous DNA, and the transgenic level variation along soya bean protein preparation were monitored systematically regarding the labelling and traceability of RRS. The results showed that ethanol extraction had more negative effect on DNA integrity during soya bean protein preparation than dehulling, defatting, alkali dissolution and acid precipitation. The RRS proportions in the final SPC and SPI products were enhanced from 0.91%–10.06% to 1.19%–11.41% and 1.12%–10.10%, respectively. These findings will be of benefit to GMO detection in processed food and provide new insights into the implementation of GMO labelling systems.

Data accessibility. Our data are deposited at the Dryad Digital Repository: https://doi.org/10.5061/dryad.j3tx95xb2 [31].
Authors' contributions. Y.D. and F.C. designed the study. Y.D. and C.C. prepared all samples for analysis. Y.D. and K.L. collected and analysed the data. Y.D. wrote the manuscript. All authors gave final approval for publication.
Competing interests. The authors declare no competing interests.
Funding. This research was supported by the National Natural Science Foundation of China (grant no. 21676073).
Acknowledgements. We thank Yong Guo for the assistance with experimental materials.

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
