## [Reviewer comments · Royal Society Open Science]

Review History

RSOS-201147.R0 (Original submission)

Review form: Reviewer 1

Is the manuscript scientifically sound in its present form?

Yes

Are the interpretations and conclusions justified by the results?

Yes

Is the language acceptable?

Yes

Do you have any ethical concerns with this paper?

No

Have you any concerns about statistical analyses in this paper?

No

Recommendation?

Accept with minor revision (please list in comments)

Comments to the Author(s)

- The supplementary table and figures are not included in the manuscript.
- Please include the primer sequences of the genes studied.

Review form: Reviewer 2**Is the manuscript scientifically sound in its present form?**

Yes

Are the interpretations and conclusions justified by the results?

Yes

Is the language acceptable?

Yes

Do you have any ethical concerns with this paper?

No

Have you any concerns about statistical analyses in this paper?

No

Recommendation?

Accept with minor revision (please list in comments)

Comments to the Author(s)

The manuscript reported the investigation of genetically modified soybean during protein concentration and isolate preparations. The study needs some minor revisions:

-In Fig. 1 and 2, please add the detailed conditions, e.g. what pH values and what compounds (NaOH, HCl etc.?) are used for alkali and acid conditions? How long does it take for each step? What volume etc.?

-Please label all ladders for the agarose gel images

-The authors miss the controls made of only GTS 40-3-2 and conventional beans.

-Are there any fragmentations on the total DNA? Is it possible that the authors run a gel to check?

Decision letter (RSOS-201147.R0)

Dear Dr Du:

Title: Monitoring and traceability of genetically modified soybean event GTS 40-3-2 during soybean protein concentrate and isolate preparation
Manuscript ID: RSOS-201147

Thank you for submitting the above manuscript to Royal Society Open Science. On behalf of the Editors and the Royal Society of Chemistry, I am pleased to inform you that your manuscript will be accepted for publication in Royal Society Open Science subject to minor revision in accordance with the referee suggestions. Please find the reviewers' comments at the end of this email.

The reviewers and handling editors have recommended publication, but also suggest some minor revisions to your manuscript. Therefore, I invite you to respond to the comments and revise your manuscript.

Because the schedule for publication is very tight, it is a condition of publication that you submit the revised version of your manuscript before 13-Aug-2020. Please note that the revision deadline will expire at 00.00am on this date. If you do not think you will be able to meet this date please let me know immediately.

Kind regards,
Dr Laura Smith
Publishing Editor, Journals

RSC Associate Editor:
Comments to the Author:
(There are no comments.)

RSC Subject Editor:
Comments to the Author:
(There are no comments.)

Reviewer comments to Author:
Reviewer: 1

Comments to the Author(s)
- The supplementary table and figures are not included in the manuscript.
- Please include the primer sequences of the genes studied.

Reviewer: 2

Comments to the Author(s)
The manuscript reported the investigation of genetically modified soybean during protein concentration and isolate preparations. The study needs some minor revisions:

-In Fig. 1 and 2, please add the detailed conditions, e.g. what pH values and what compounds (NaOH, HCl etc.?) are used for alkali and acid conditions? How long does it take for each step? What volume etc.?

-Please label all ladders for the agarose gel images

-The authors miss the controls made of only GTS 40-3-2 and conventional beams.

-Are there any fragmentations on the total DNA? Is it possible that the authors run a gel to check?

Author's Response to Decision Letter for (RSOS-201147.R0)

See Appendix A.

RSOS-201147.R1 (Revision)

Review form: Reviewer 2

Is the manuscript scientifically sound in its present form?

Yes

Are the interpretations and conclusions justified by the results?

Yes

Is the language acceptable?

Yes

Do you have any ethical concerns with this paper?

No

Have you any concerns about statistical analyses in this paper?

No

Recommendation?

Accept as is

Comments to the Author(s)

The authors have carefully addressed the previous comments and the review does not have further comments

Decision letter (RSOS-201147.R1)

Dear Dr Du:

Title: Monitoring and traceability of genetically modified soybean event GTS 40-3-2 during soybean protein concentrate and isolate preparation

Manuscript ID: RSOS-201147.R1

It is a pleasure to accept your manuscript in its current form for publication in Royal Society Open Science. The chemistry content of Royal Society Open Science is published in collaboration with the Royal Society of Chemistry.

RSC Associate Editor:
Comments to the Author:
(There are no comments.)

RSC Subject Editor:
Comments to the Author:
(There are no comments.)

Reviewer(s)' Comments to Author:
Reviewer: 2

Comments to the Author(s)
The authors have carefully addressed the previous comments and the review does not have further comments

Appendix A

Manuscript Draft

Manuscript ID: RSOS-201147R1

Title: Monitoring and traceability of genetically modified soybean event GTS 40-3-2 during soybean protein concentrate and isolate preparation

Article Type: Research Article

Keywords: Genetically modified soybean;

Real-time PCR;

Traceability;

Soybean protein concentrate;

Soybean protein isolate

Abstract: To evaluate DNA fragmentation and GMO quantification during soybean protein concentrate and isolate preparation, genetically modified soybean event GTS 40-3-2 (RRS) was blended with conventional soybeans at mass percentages of 0.9%, 2%, 3%, 5%, and 10%. Qualitative PCR and real-time PCR were used to monitor the taxon-specific *lectin* and exogenous *cp4 epsps* target levels in all of the main products and by-products, which has practical significance for RRS labelling threshold and traceability. Along the preparation chain, the majority of DNA was distributed in main products, and the DNA degradation was noticed. From a holistic perspective, the *lectin* target degraded more than *cp4 epsps* target during both of the two soybean proteins preparations. Therefore, the transgenic contents in the final protein products were higher than the actual mass percentages of RRS in raw materials. Our results are beneficial to the improvement of GMO labelling legislation and the protection of consumer rights.

<Journal Name>: Royal Society Open Science

<Manuscript ID>: RSOS-201147R1

<Manuscript Title>: Monitoring and traceability of genetically modified soybean event GTS 40-3-2 during soybean protein concentrate and isolate preparation

Dear Editor Dr. Laura Smith:

Thank you very much for your kind information. We highly appreciate the meaningful comments from editor and reviewers concerning our manuscript entitled “**Monitoring and traceability of genetically modified soybean event GTS 40-3-2 during soybean protein concentrate and isolate preparation**”. Those comments are all valuable and very helpful for revising and improving our manuscript, as well as the important guiding significance to our researches. We have carefully considered the comments and have tried our best to revise the manuscript according to the comments.

Enclosed please find the revised manuscript, which we would like to submit for your kind consideration. A point-by-point list of our response (highlighted in blue) to the Reviewers’ and Editor’s comments have been attached.

We hope this manuscript is suitable for “**Royal Society Open Science**”. We would like to express our great appreciation to you and the reviewers for comments on our manuscript. Looking forward to hearing from you.

Thank you and best regards,

Sincerely yours,

Fusheng Chen, Ph.D.

Professor

Henan University of Technology

Zhengzhou, 450001, P.R. China

E-mail: fushengc@haut.edu.cn

Response to the Reviewers' and Editor's comments

Reviewers' comments:

Reviewer #1:

Comment 1: The supplementary table and figures are not included in the manuscript.

Authors' response: We appreciate the reviewer very much for the careful comments. Maybe because the supplementary table and figures were uploaded in the form of 'Electronic supplementary material (ESM)' in the submission system, they were not presented in the manuscript. Thanks again for the kind suggestions.

Comment 2: Please include the primer sequences of the genes studied.

Authors' response: Thanks for reviewer's insightful suggestions. The primer sequences of the genes studied are shown as follows. We have included them in Table S1 as a supplementary material. Please refer to Table S1 for more details. Thanks again for the kind suggestions.

Table S1 Primers and amplification conditions used in qualitative PCR

Purpose	Product size (bp)	Location	Reference	Primer sequences	Annealing temperature (°C)	Extension time (s)
lectin (K00821)	60	1273-1332	[1]	F1 5'-TCGCCGCTTCCTTCAACTT-3' R1 5'-GCCCATCTGCAAGCCTTT-3'	52	9
	201	1242-1442	[2]	F2 5'-TGGGACAAAGAAACCGGTAG-3' R2 5'-GTCAAACCTAACAGCGACGA-3'	55	15
	414	1099-1512	[3]	F3 5'-TGCCGAAGCAACCAACATGATCCT-3' R3 5'-TGATGGATCTGATAGAATTGACGTT-3'	55	30
	836	927-1762	[4]	F4 5'-GACTCCCCATGCATCACAGT-3' R4 5'-GGCAAATTGGAAGCAAAAAGA-3'	60	45
	1487	303-1789	[5]	F5 5'-TCTTTTAGTCCATGTATTCT-3' R5 5'-AAAGGATCAATGTTACTGCT-3'	54	90
	70	1494-1563	[5]	F1 5'-ATATCCGATTCTCGCTGTCGC-3' R1 5'-GAGTTCTTCCAGACCGTTCAT-3'	52	9
	256	1336-1591	[6]	F2 5'-ACCGCCTCATCCTGACGCT-3' R2 5'-CCGAGAGGCGGTCGCTTTCC-3'	59.8	21
	408	1371-1778	[4]	F3 5'-CGACATCGAAGTCATCAACC-3' R3 5'-GTGACAGGGTTTTCCGACAC-3'	55	30
	807	1028-1834	[4]	F4 5'-CCTCCGCACAGGTGAAGT-3' R4 5'-CCATCAGGTCCATGAACTCC-3'	60	45
	1102	539-1640	[2]	F5 5'-CCGCAACCGCCCGCAAATCCTCT-3' R5 5'-TCGCCCTCATCGCAATCCACGCC-3'	63.1	72
1339	539-1877	[7]	F6 5'-CCGCAACCGCCCGCAAATCCTCT-3' R6 5'-GCAGCCTTCGTATCGGAGAGTTC-3'	59.3	86	
1512	670-2181	[4]	F7 5'-GGCGAGGACGTCATCAATAC-3' R7 5'-TCGATCCCCGATCTAGTAACA-3'	54	90	

Reviewer #2:

The manuscript reported the investigation of genetically modified soybean during protein concentration and isolate preparations. The study needs some minor revisions:

Authors' response: We appreciate the reviewer very much for the constructive comments and useful guidances on our work. According to your suggestions, we have addressed the following points seriously and amended the relevant parts in the revised manuscript. A list of response to each point was raised below carefully. Please refer to the responses to Comment 1-4 for more details.

Comment 1: In Fig. 1 and 2, please add the detailed conditions, e.g. what pH values and what compounds (NaOH, HCl etc.?) are used for alkali and acid conditions? How long does it take for each step? What volume etc.?

Authors' response: Thanks very much for reviewer's kind suggestions. We have added the detailed conditions in Fig.1 and 2. Please refer to Fig.1 and 2 in the revised manuscript for details. Thanks again for the useful comments.

Comment 2: Please label all ladders for the agarose gel images.

Authors' response: Thanks for reviewer's precise comments and kind reminders. We have labelled all ladders for the agarose gel images. Please refer to Fig.3-6 in the revised manuscript for details. Thanks again for the useful comments.

Comment 3: The authors miss the controls made of only GTS 40-3-2 and conventional beans.

Authors' response: Thanks for reviewer's useful and insightful suggestions. We agree with the reviewer that the samples made of pure GTS 40-3-2 and conventional soybeans should be detected, which have been investigated previously by our team [8]. In our previous research, the mass variations of genomic DNA and length distributions of DNA fragments in pure GTS 40-3-2 and conventional soybeans and the variations in transgenic contents during soybean protein concentrate (SPC) and soybean protein isolate (SPI) preparation were monitored. The results showed that the distribution of DNA in the samples differed, and the *lectin* and *cp4 epsps* targets degraded to some extent along the preparation chains. Meanwhile, the DNA distribution and degradation thereby affected GMO quantification.

On the basis of our previous research, genetically modified soybeans with RRS mass percentages of

0.9%, 2%, 3%, 5%, and 10% were selected to further clarify the variations of RRS proportions during the SPC and SPI preparation started from raw soybean materials with different transgenic contents. The RRS mass percentages (0.9%, 2%, 3%, 5%, and 10%) used in this research were chosen according to the labelling rules worldwide. For example, the rules in Japan, Thailand, Taiwan, South Africa, and Indonesia state that any food product should be clearly labelled if it contains 5% or more GMO components [9]. The corresponding threshold values are 3% in South Korea [10] and 0.9% in the European Union and Russia [11].

Therefore, the samples made of pure GTS 40-3-2 and conventional beans were not included in this research. Thanks again for the kind suggestion.

Comment 4: Are there any fragmentations on the total DNA? Is it possible that the authors run a gel to check?

Authors' response: Thanks for reviewer's precise and careful comments. We have run a gel to check whether there are any fragmentations on the total DNA. The gel electrophoresis results are shown below.

Fig. 1 Agarose gel electrophoresis analysis of DNA extracted from samples collected during SPC and SPI preparation. Lane (M) λ DNA/Hind III Marker, (1) raw soybean material, (2) soybean hull, (3) soybean kernel, (4) defatted soybean kernel, (5) SPC precipitate 1, (6) SPC precipitate 2, (7) SPC precipitate 3, (8) SPC product, (9) SPC supernatant 1, (10) SPC supernatant 2, (11) SPC supernatant 3, (12) SPC supernatant 4, (13) SPI supernatant 1, (14) SPI product, (15) SPI precipitate 1, (16) SPI supernatant 2, (17) soybean oil.

As revealed in Fig. 1, the integrities of genomic DNAs extracted from raw soybean material (1), soybean kernel (3), and defatted soybean kernel (4) were high. While, the bands of genomic DNAs

extracted from SPC precipitate 1 (5), SPC precipitate 2 (6), SPC precipitate 3 (7), and SPC product (8), as well as those of genomic DNAs extracted from SPI supernatant 1 (13) and SPI product (14) were weakened and diffused gradually. Furthermore, maybe due to the low content of genomic DNA extracted from SPC supernatant 1 (9), SPC supernatant 2 (10), SPC supernatant 3 (11), SPC supernatant 4 (12), SPI supernatant 2 (16), and soybean oil (17), the bands of lane 9-12 and 16-17 were too weak to be detected.

Therefore, it was confirmed that fragmentations of the total DNA happened during SPC and SPI preparations. Thanks again for the kind comments.

References

1. He J, Xu W, Shang Y, Zhu P, Mei X, Tian W, Huang K. 2013 Development and optimization of an efficient method to detect the authenticity of edible oils. *Food Control*. **31**, 71-79. (doi: 10.1016/j.foodcont.2012.07.001)
2. Bergerová E, Hrnčířová Z, Stankovská M, Lopařovská M, Siekel P. 2010 Effect of thermal treatment on the amplification and quantification of transgenic and non-transgenic soybean and maize DNA. *Food Anal. Method*. **3**, 211-218. (doi: 10.1007/s12161-009-9115-y)
3. Tian F, Guan Q, Wang X, Teng D, Wang J. 2014 Influence of different processing treatments on the detectability of nucleic acid and protein targets in transgenic soybean meal. *Appl. Biochem. Biotech*. **172**, 3686-3700. (doi: 10.1007/s12010-014-0760-2)
4. Chen Y, Wang Y, Ge Y, Xu B. 2005 Degradation of endogenous and exogenous genes of roundup-ready soybean during food processing. *J. Agr. Food Chem*. **53**, 10239-10243. (doi: 10.1021/jf0519820)
5. Zhang X, Chen F, Zhang L, Xin Y. 2019 Distribution of endogenous and exogenous genes in the process of aqueous enzymatic extraction of genetically modified soybean oil. *Food Research and Development*. **40**, 1-6. (doi: 10.3969/j.issn.1005-6521.2019.04.001)
6. Datukishvili N, Kutateladze T, Gabriadze I, Bitskinashvili K, Vishnepolsky B. 2015 New multiplex PCR methods for rapid screening of genetically modified organisms in foods. *Front. Microbiol*. **6**, 757. (doi: 10.3389/fmicb.2015.00757)

7. Bauer T, Weller P, Hammes WP, Hertel C. 2003 The effect of processing parameters on DNA degradation in food. *Eur. Food Res. Technol.* **217**, 338-343. (doi: 10.1007/s00217-003-0743-y)
8. Du Y, Chen F, Bu G, Zhang L. 2021 Distribution and degradation of DNA from non-genetically and genetically modified soybean (Roundup Ready): Impact of soybean protein concentrate and soybean protein isolate preparation. *Food Chem.* **335**, 127582. (doi: 10.1016/j.foodchem.2020.127582)
9. Nikolić Z, Petrović G, Panković D, Ignjatov M, Marinković D, Stojanović M, Đorđević V. 2017 Threshold level and traceability of Roundup Ready® soybeans in Tofu production. *Food Technol. Biotech.* **55**, 439-444. (doi: 10.17113/ftb.55.04.17.5192)
10. Kim JH, Song JY, Hong Y, Kim HY. 2016 Monitoring of genetically modified soybean events in sausage products in South Korea. *Food Control.* **67**, 63-67. (doi: 10.1016/j.foodcont.2016.02.041)
11. Plácido A, Pereira C, Guedes A, Barroso MF, Miranda-Castro R, de-los-Santos-Álvarez N, Delerue-Matos C. 2018 Electrochemical genoassays on gold-coated magnetic nanoparticles to quantify genetically modified organisms (GMOs) in food and feed as GMO percentage. *Biosens. Bioelectron.* **110**, 147-154. (doi: 10.1016/j.bios.2018.03.042)